# Effects of Home-Based Exercise Training on Cardiac Autonomic Neuropathy and Metabolic Profile in Diabetic Hemodialysis Patients

**DOI:** 10.3390/life13010232

**Published:** 2023-01-13

**Authors:** Vassiliki Michou, Vassilios Liakopoulos, Stefanos Roumeliotis, Athanasios Roumeliotis, Maria Anifanti, Georgios Tsamos, Aikaterini Papagianni, Pantelis Zempekakis, Asterios Deligiannis, Evangelia Kouidi

**Affiliations:** 1Sports Medicine Laboratory, School of Physical Education & Sport Science, Aristotle University, 57001 Thessaloniki, Greece; 2Division of Nephrology and Hypertension, 1st Department of Internal Medicine, Medical School, AHEPA Hospital, Aristotle University, 57001 Thessaloniki, Greece; 3Laboratory of Hygiene, Department of Internal Medicine, Agios Dimitrios Hospital, Aristotle University, 57001 Thessaloniki, Greece; 4Department of Nephrology, Hippokration Hospital, Aristotle University, 57001 Thessaloniki, Greece

**Keywords:** diabetic kidney disease, hemodialysis, cardiac autonomic neuropathy, exercise, physical activity, metabolic profile

## Abstract

Background: This study aimed to investigate the effects of a home-based exercise training program on Cardiac Autonomic Neuropathy (CAN) and metabolic profile in Diabetic Kidney Disease (DKD) patients undergoing maintenance hemodialysis (HD). Method: Twenty-eight DKD patients undergoing hemodialysis were randomly assigned into two groups. The exercise (EX) group followed a 6-month combined exercise training program at home, while the control (CO) group remained untrained. All participants at baseline and the end of the study underwent cardiopulmonary exercise testing (CPET), biochemical tests for glucose and lipid profile, and 24-h electrocardiographic monitoring for heart rate variability (HRV) analysis and heart rate turbulence (HRT). Results: At the end of the study, compared to the CO, the EX group showed a significant increase in serum high-density lipoprotein (HDL) by 27.7% (*p* = 0.01), peak oxygen uptake (VO_2_peak) by 9.3% (*p* < 0.05), the standard deviation of R-R intervals (SDNN) by 34.3% (*p* = 0.03), percentage of successive RR intervals higher than 50ms (pNN50) by 51.1% (*p* = 0.02), turbulence slope (TS) index by 18.4% (*p* = 0.01), and decrease in (glycated hemoglobin) HbA1c by 12.5% (*p* = 0.04) and low-frequency power LF (ms^2^) by 29.7% (*p* = 0.01). Linear regression analysis after training showed that VO_2_peak was correlated with SDNN (r = 0.55, *p* = 0.03) and HF (r = 0.72, *p* = 0.02). Multiple regression analysis indicated that the improvement of sympathovagal balance and aerobic capacity depended on patients’ participation in exercise training. Conclusion: In conclusion, a 6-month home-based mixed-type exercise program can improve cardiac autonomic function and metabolic profile in DKD patients on HD.

## 1. Introduction

Cardiac Autonomic Neuropathy (CAN) is the most frequent complication of Diabetes Mellitus (DM). CAN is a strong predictor of cardiovascular events, reduces the quality of life, [1] and increases the risk for mortality in these patients up to 5 times more [2] The prevalence of CAN varies and ranges on average from 2% to 91% in type 1 DM (DM-1) and from 25% to 75% in type 2 DM (DM-2) [3,4] and has been suggested to promote the progression of kidney disease. Diabetic Kidney Disease (DKD) is the most common cause of Chronic Kidney Disease (CKD), which increases the risk of arrhythmias and sudden cardiac death [5,6]. However, mechanisms linking CAN, DKD, and cardiovascular mortality are not yet fully elucidated [6].

Since CAN is closely related to reduced work tolerance and inappropriate heart rate, blood pressure and cardiac output in response to exercise, patients with increased CAN are having difficulties in performing their daily activities. On the other hand, increased physical activity is highly recommended for these populations [7,8,9,10], to ameliorate the CV burden by increasing heart rate variability (HRV) and improving left ventricular (LV) function (especially for heart patients) [11,12]. Pathophysiological mechanisms of exercise on cardiac autonomic function, functional capacity, lipid, and glucose profile in DKD patients are shown in Figure 1.

Even though the effects of exercise have been thoroughly investigated in CKD and DM patients, there are no clinical trials or cohort studies regarding the impact of training on cardiac autonomic dysfunction in DKD patients. Therefore, the present study aimed to investigate the effects of a long-term, home-based, combined exercise program on CAN and metabolic profile in hemodialysis (HD) patients with DKD.

## 2. Materials and Methods

### 2.1. Patients

Initially, 60 DKD patients undergoing maintenance HD with diagnosed CAN were screened for eligibility. Forty-one patients who met the inclusion criteria and volunteered to participate in our study were randomly assigned to either EX or CO group. All patients were recruited from public and private dialysis units of the Prefecture of Thessaloniki in Greece. Inclusion criteria included: age < 80-year-old patients undergoing hemodialysis treatment and, over six months vintage of DM-2. Exclusion criteria included age > 80 years old, history of clinical coronary heart disease (CHD) within the previous six months, severe musculoskeletal problems that may limit the patient’s participation in this study, no compliance with diabetes medication, receiving medication that affects the ANS and previous involvement in an exercise training program.

### 2.2. Study Design

Initially, patients with DKD who met the inclusion criteria underwent a review of the medical history, clinical examination, a 12-lead [electrocardiogram (ECG)], 24-h electrocardiographic monitoring for heart rate variability (HRV) and heart rate turbulence (HRT) analysis, blood sampling and cardiopulmonary exercise testing (CPET) for their functional capacity estimation, at the Sports Medicine Laboratory of the Department of Physical Education and Sports Science of the Aristotle University of Thessaloniki, Greece.

After baseline measurements, patients were randomly assigned to either an exercise group (EX group) or a control group (CO group). For the randomization process, the www.randomizer.org (accessed on 30 January 2020) website was used. Patients in the CO group were asked to continue their daily routine and avoid participating in any structured exercise program. All tests were conducted by the same researcher, blinded to group allocation. Furthermore, all patients’ medications were asked to remain unchanged during the study period to exclude medication effects (Table 1). At the end of the 6-month study, all baseline assessments were repeated. This randomized controlled trial protocol was approved by the Ethics Committee of the Aristotle University of Thessaloniki (Protocol number: 117461/2019). All participants received all the necessary study information before enrollment and provided written informed consent. The clinical trial started in February 2020 and ended in June 2020.

### 2.3. Charlson Comorbidity Index

The original version of the Charlson Comorbidity Index (CCI), which is a worldwide indicator that predicts long-term mortality for patients with multiple comorbidities, was used in both groups. The CCI is a 19-item scale corresponding to different medical comorbid conditions combined with age. According to Charlson et al. [13] and Fraccaro et al. [14], the higher the CCI, the higher the mortality over ten years [15]. Huang et al. [16] showed that in DKD patients with DM-2 during a 5-year follow-up period, the more the mortality rate increased, the higher the CCI was.

### 2.4. Blood Analysis Assessment

Blood samples were taken after a 12-h fast, at least five days before the exercise training sessions, began and 24–48 h after the end of the 6-month exercise protocol. Blood analysis through biochemical auto-analyzer devices included determination of hematocrit by photometric method, hemoglobin by computational method, serum electrolytes (potassium, sodium, calcium, phosphorus, magnesium) by ion-selective electrode method, fasting plasma glucose (FBG), serum triglycerides (TG) and glycated hemoglobin (HbA1c) by enzymatic colorimetric method and serum high-density lipoprotein (HDL), low-density lipoprotein (LDL) and total cholesterol (TC) by enzymatic method.

### 2.5. Cardiopulmonary Exercise Testing

A symptom-limited cardiopulmonary exercise testing (CPET) on a treadmill using the Bruce protocol [17] was used to assess DKD patients’ functional capacity. Patients underwent CPET during morning hours (between 9:00 and 11:00 am. The electrocardiogram (GE Medical Systems, Milwaukee, WI, USA) was continuously monitored, while blood pressure was measured at the end of each stage. In addition, the Med Graphics Breeze Suite CPX Ultima (Medical Graphics Corp, Milwaukee, WI, USA) measured breath-by-breath gas exchange. Gas indicators that were analyzed were the peak oxygen uptake (VO_2_peak), pulmonary ventilation (VE), ventilatory equivalents for oxygen (VE/VO_2_), carbon dioxide (VE/VCO_2_), and the ratio between VO_2_ and maximum HR (VO_2_/HRmax). The endpoint was set as the respiratory exchange ratio ≥ 1.10 or the oxygen plateau during maximal exercise.

### 2.6. 24-H Electrocardiographic Monitoring

DKD patients underwent a 24-recording of the ECG to evaluate the cardiac autonomic function using a small portable 3-channel (with seven electrodes) ECG Holter device (SEER 1000, GE Healthcare, Chalfont St Giles, UK). In addition, patients were asked not to consume alcohol for at least 12 h and to refrain from vigorous physical activity for at least 24 h before the scheduled measurement. Data were stored and analyzed using the CardioDay software (GE Healthcare, Chalfont St Giles, UK) to estimate Heart Rate Variability (HRV) indices in the Time and Frequency Domain and Heart Rate Turbulence (HRT). Concerning HRV time domain analysis, four variables were evaluated [1]: the standard deviation of R-R (the time intervals between two successive heart beats) intervals (SDNN) [2], the standard deviation of R-R intervals calculated every 5 min (SDANN) [3], the square root of the mean sum of the squares of the differences between consecutive intervals R-R (rMSSD) and [4] the percentage of successive RR intervals higher than 50 ms (pNN50). Similarly, from the HRV frequency domain analysis, the following five indices were estimated [1]: the total frequency power (TP) [2], the very low-frequency power (VLF) (<0.003–0.04 Hz) [3], the low-frequency power (LF) (0.04–0.15 Hz) [4], the high-frequency power (HF) (0.15–0.4 Hz) and [5] the frequency ratio (LF/HF). In addition, the HRT indicators analyzed were the turbulence onset (TO) index and the turbulence slope (TS) index.

### 2.7. Exercise Training Program

The EX group followed a 6-month home-based exercise training program on the non-dialysis days. The exercise program consisted of 3 combined (aerobic and strengthening) exercise sessions per week. Each exercise session was 60–90 min long, of moderate intensity, i.e., at 50–70% of predicted VO_2_peak achieved during CPET, and started with a 10-min warm-up and ended with 10-min recovery exercises (upper and lower extremity stretches). In addition, during the first week, all patients received an information exercise booklet and had three familiarization sessions with accredited physical education teachers with expertise in exercise rehabilitation for patients with chronic diseases.

Walking or cycling on a stationary bicycle was recommended as aerobic exercise. Patients were informed to start exercise initially for 15 min with a consequent gradual increase of time by 5 min every two weeks, reaching 40 min in the last two weeks before the end of the program. The strengthening exercise included six dynamic muscle strengthening exercises, performed in 2 sets of 8–10 repetitions (with a 1-min passive break between the sets) in a progressive sequence from sitting to standing. Initially, patients started with strengthening exercises for the upper and lower limbs using only their body weight. Afterward, they performed the same limb exercises using rubber bands, balls, and dumbbells (1 kg). These exercises included 3 phases: (a) in sitting position, two sets (8–10 repetitions) of upper limb strengthening exercise with balls and two sets with the 1 kg dumbbells, (b) in sitting position strengthening exercises (2 sets, 8–10 repetitions) by lacing the rubber bands on feet and tie them to the bottom of the bed or chair and (c) in standing position, with hands in the middle of the body and by placing the dumbbells on feet, moving the legs back and forth, right and left of the torso.

The 6-month home-based exercise training program was individualized to ensure each patient’s autonomy. The researcher monitored the progress and adherence to the program via telephone at the end of each week and by a monthly home visit (wherever possible) to record improvements and provide modifications to the exercise program. Moreover, DKD patients were asked to fill in individual diaries, describing the type, frequency, and duration of each exercise session or missing sessions for any reason. The EX group included patients in the analysis, and they participated in at least 85% of the scheduled sessions.

DKD patients, after proper training, were also asked to measure blood glucose, blood pressure, and HR levels at least 20 min before starting the exercise session and note their measurements in their diaries. In case of high (over 130 mg/dL) or low (below 70 mg/dL) blood glucose levels, patients were advised not to start exercising. Patients were also advised to stop exercise in case of illness (i.e., dizziness, visual disturbance, or severe shortness of breath).

### 2.8. Statistical Analysis

The IBM Statistical Package for Social Sciences (IBM Corp. Released 2020. IBM SPSS Statistics for Windows, Version 27.0. Armonk, NY, USA: IBM Corp) was used for statistical analysis. The Kolmogorov-Smirnov test was used to assess the normal distribution of variables. Mean differences within time and between the two groups were analyzed using two-way ANOVA with repeated measures. The differences between the EX and CO groups regarding changes in the examined parameters were analyzed with the *t*-test for independent samples. Intra- and inter-observer variability were determined by intraclass correlation coefficient (CCI) and 95% confidence intervals (Cl). Linear regression was used to study the association between variables that revealed statistically significant changes over time, while multiple linear regression analysis was performed to evaluate the impact of confounding factors on results. Data were expressed as mean ± standard deviation for normally distributed variables. The significance level for accepting or not having a statistically significant difference for all statistical tests was set at *p* < 0.05.

## 3. Results

### 3.1. Patients’ Characteristics

Six patients from the EX group and seven from the CO group withdrew during the follow-up period; therefore, 28 patients completed the study (Figure 2). According to per-protocol analysis, patients of the EX group performed 95 ± 2% of the scheduled sessions according to their diaries. During the six months, none of the patients showed any exercise-induced cardiovascular or musculoskeletal complications. The demographic and clinical characteristics of DKD patients are shown in Table 2.

### 3.2. Blood Analysis

At baseline, there was no statistically significant difference in any blood test indicator between the EX and CO group. After the 6-month exercise program, the EX group showed a significant decrease in FPG by 15.9% (*p* < 0.05), TC by 6.3% (*p* = 0.04), TG by 10.1% (*p* = 0.01) and HbA1c by 10.2% (*p* < 0.05), while an increase in HDL by 22.3% (*p* < 0.05) was also noticed. Regarding the inter-group changes between the EX and CO group at the end of the study, results for the EX group showed a favorable increase in HDL by 27.7% (*p* = 0.01) and a decrease in HbA1c by 12.5% (*p* = 0.04), compared to the CO group (Table 3).

### 3.3. Cardiorespiratory Fitness

We found that after six months, the EX group showed a significant increase in exercise time by 7.4% (*p* = 0.03), METs by 4.8% (*p* < 0.05), VO_2_peak by 9.8% (*p* < 0.05) and exercise HR by 4.7% (*p* < 0.05), while a significant decrease was noticed in VE/VO_2_max by 5.6% (*p* < 0.05) and VE/VCO_2_max by 4.1% (*p* = 0.01) (Table 3). There were no statistically significant differences between groups at baseline. After 6 months, the EX group showed a statistically significant increase in METs by 5.4% (*p* = 0.04) and VO_2_peak by 9.3% (*p* < 0.05), while lower values were observed in resting HR by 3.0% (*p* < 0.05), resting SBP by 2.4% (*p* = 0.03), exercise SBP by 4.7% (*p* < 0.05) and DBP by 3.6% (*p* = 0.01), compared to the CO group. Patients allocated to the CO group failed to demonstrate significant improvements in the above indices (Table 4).

### 3.4. 24-H Holter Monitoring

EX group results from the HRV analysis showed significant improvements in TP (increase by 12.5%, *p* = 0.04), SDNN (increase by 24.5%, *p* < 0.05), SDANN (increase by 19.0%, *p* = 0.02), rMSSD (increase by 21.9%, *p* < 0.05), pNN50 (increase by 29.0%, *p* = 0.04), LF [ms^2^ (decrease by 30.8%, *p* = 0.03)], HF [ms^2^ (increase by 29.1%, *p* = 0.01)], LF [n.u. (decrease by 22.7%, *p* = 0.02)] and TS (increase by 18.4%, *p* = 0.01) at the end of the study (Table 5). In contrast, there was no statistically significant difference in any HRV index in the CO group after six months. Moreover, inter-group changes at the end of the study showed that EX group statistically increased SDNN by 34.3% (*p* = 0.03), rMSSD by 21.5% (*p* = 0.02) and pNN50 by 51.7% (*p* = 0.02), and decreased LF (ms^2^) by 29.7% (*p* = 0.01), compared to the CO group.

### 3.5. Linear Regression Analysis

At the end of the study, a positive linear relationship was found in the EX group between SDNN and VO_2_peak (r= 0.55, *p* = 0.03) (Figure 3), HF (ms^2^) and VO_2_peak (r = 0.72, *p* = 0.02) (Figure 3), time of exercise and SDNN (r = 0.62, *p* = 0.04) (Figure 4), while a negative linear relationship was noticed only between rMSSD and HbA1c (r= −0.70, *p* < 0.05) (Figure 5).

### 3.6. Multiple Linear Regression Analysis

Finally, at the end of the study, multiple linear regression analysis was performed to examine the relationship between sympathovagal balance, as measured with SDNN, and aerobic capacity, as measured with VO_2_peak, with a variety of independent variables. By using SDNN as the dependent variable, results showed that lower values of HbA1c (*p* = 0.02) and increased VO_2_peak (*p* < 0.05) had a statistically significant contribution to the model (Table 6). More precisely, results revealed that 72.4% of the variability observed in SDNN was explained by the regression model (R^2^ = 0.724, F = 3.505, *p* = 0.04). In addition, by using VO_2_peak, as a subordinate variable, the analysis showed that higher values of SDNN (*p* = 0.03), HF (ms^2^) (*p* < 0.05), and rMSSD (*p* = 0.03) had a significant contribution to the model (Table 7), which explained 86.3% of the total variance (R^2^ = 0.863, F = 4.706, *p* = 0.03).

## 4. Discussion

Our findings showed that a 6-month home-based exercise program has favorable effects on cardiac autonomic function and functional capacity in DKD patients undergoing HD with CAN. This 6-month exercise training program significantly improved the sympathetic and vagal nerve activity and the sympathovagal balance in our cohort.

The exact mechanisms underlying the beneficial effect of exercise on cardiac autonomic nervous system (ANS) function are not yet completely fully elucidated [18]. However, it has been shown that regular exercise training improves cardiac ANS activity through specific molecular mediators, such as nitric oxide (NO) [19] and angiotensin II, a well-known inhibitor of cardiac vagal tone [20]. Physical exercise might increase NO bioavailability and suppress the activity of angiotensin II, thus leading to increased cardiac vagal tones. Therefore, subjects undergoing regular physical exercise have significantly lower plasma renin levels than those living a sedentary lifestyle [21].

An alternative, highly reproducible measure of an individual’s ability to recruit cardiac vagal tone is heart rate recovery (HRR) after exercise [22,23,24]. In heart failure [25,26] and DM-2 [20] patients, exercise training accelerates HRR. Thus, HRR may be considered a strong indicator of improved cardiac vagal function. These references suggest that regular exercise can indeed “train“ vagal tone. Still, the potential neurophysiological mechanisms (e.g., recruitment of more vagal neurons or the more efficient transmission of nerve impulses at the ganglion level) involved in the apparent plasticity of the cardiac ANS remain unknown.

Reducing 24-h-SDNN recording is considered the “gold standard” for increased cardiovascular risk, as its values can predict morbidity and mortality [27,28]. Kleiger et al. [29], showed that patients with SDNN values less than 50ms are at increased cardiovascular risk, whereas those with SDNN values higher than 100ms have an approximately 5.3 times lower mortality risk. Pagkalos et al. [30] showed that a 6-month aerobic and resistance exercise training program, at 70–85% of the HRmax, can significantly improve most HRV indices and increase the sympathovagal balance of DM-2 patients with CAN. Regardless of the exercise type, a 3-month exercise program, 3–5 times per week, was found to improve HRV indices by enhancing the vagal nerve activity and reducing the sympathetic activity in patients with Type 2 Diabetic Neuropathy [31].

Results of the present study revealed significant improvements in the indices reflecting both vagal and sympathetic activity, leading to enhanced sympathovagal balance after exercise training. Deligiannis et al. [32] investigated the effect of a 6-month exercise program on HRV indices in HD patients and found favorable improvements in the HRV triangular index. In addition, the VO_2_peak increased significantly after the training, while a significant correlation between HRV index and VO_2_peak was also found at the end of the study. Likewise, Kouidi et al. [33,34] highlighted substantial improvements in cardiac ANS function after a long-term supervised exercise program during the HD sessions. In disagreement with these results, in a study with a similar study design, Morais et al. [35] showed that a 3-month aerobic intradialytic exercise did not improve cardiac ANS activity. Similarly, Huppertz et al. [11], examining the effect of lifestyle change and participation of 113 patients in a gym and home exercise program, did not notice significant changes in cardiac ANS activity.

Moreover, the present study revealed a statistically significant increase in the TS index of HRT (by 18.4%) for the EX group after training. The present study is the first to evaluate the change in HRT, combined with HRV indices, after a long-term exercise program in DKD patients. Few studies have evaluated the diagnostic value of HRT on CAN in diabetic patients, although it has a significant diagnostic value so far [36]. HRT is considered an indicator of vagal activity and an independent factor of total mortality [37]. TO and TS values correlate highly with HRV parameters, such as SDNN and rMSSD [38]. Lin et al. [39] evaluated and compared the Ewing test, HRV, and HRT indices to diagnose CAN in 90 diabetic patients. They found that using both HRV and HRT analysis, the CAN rate was 56.6% and 52.2%, respectively. Combining the TS index with SDNN, the diagnostic sensitivity for CAN can increase up to 98.0%. Accordingly, Disertori et al. [40] reported that HRV strongly predicts sudden cardiac death and arrhythmic events, particularly in patients with an ejection fraction >30% after acute myocardial infarction.

Our study also revealed a statistically significant increase in VO_2_peak, METs, and exercise time after a 6-month home-based combined exercise program. Several studies have shown significant improvements in cardiorespiratory efficiency and functional capacity in HD or DM patients after a home-based exercise training program. For example, Myers et al. [41] observed that a supervised 12-week home exercise program in HD patients increased the 6-min walking test performance (6MWT), VO_2_peak, and exercise time. Similarly, Baggetta et al. [42] showed favorable effects of a 6-month, low-intensity exercise program at home in a total of 115 hemodialysis patients (37% with DM) in the EXCITE study. In addition, results from our study revealed positive correlations between VO_2_peak and SDNN and HF after exercise training, indicating that the cardiorespiratory efficiency levels affect cardiac ANS function. Although many studies focus primarily on the beneficial effects of intradialytic exercise, there is no significant difference in the outcomes achieved by home-based exercise training programs. In fact, practice at home has advantages since it can increase patients’ autonomy and subsequent compliance with exercise training [43].

Furthermore, our study revealed favorable intra-group improvements for glucose and lipid indices after six months, while a strong negative correlation between HbA1c and rMSSD was also noticed. Risk factors that increase cardiovascular mortality in DM-2 patients with CAN include dyslipidemia, duration of DM, abnormal glucose tolerance, hypertension, and obesity. Exercise programs could offset some of these risks. Mild to moderate physical activity increases lipolysis of triacylglycerols, induced by the increased response to catecholamines, contributing positively to the reduction of insulin resistance [44]. However, in HD patients, the results of previous studies are controversial. In the study of Sanavi et al. [45], after an 8-week intradialytic mixed-type exercise program, improvements were observed for CRP and creatinine levels but not for lipids. In contrast, Song and Sohng [46], after a 12-week progressive resistance training program, observed an inter-group reduction in TC and TG. Likewise, Torres et al. [47] found a significant decrease in LDL and TG levels after a 3-month exercise training program. According to Meher and Panda [48], CAN could strongly correlate with high HbA1c levels, as uncontrollable hyperglycemia could be a possible explanation for CAN deterioration. Pagkalos et al. [30] have also noticed a similar correlation between the improvements in HbA1c levels and vagal activity, as indicated with pNN50 and rMSSD, in DM-2 patients. Additionally, our results are in agreement with previous studies [49,50]. However, even though the strong correlation between CAN and poor glycemic control is well-established, more studies are needed to examine these correlations in the DKD population.

The present study has strengths and limitations that need to be acknowledged. It is the first randomized controlled trial that evaluated the cardiac ANS effects of a long-term, home-based combined exercise program in DKD patients undergoing maintenance HD. These results are noteworthy, as cardiovascular disease is responsible for approximately 40% of deaths in ESKD patients [51], and the prevalence of DM-2, according to global estimations, will increase by 3% to 6% at the end of 2025, with approximately 3 million patients with DM-2 [52]. On the other hand, a limitation of this study is its small sample size, mainly due to the difficulties of recruiting patients for long-term exercise training studies. Secondly, even though patients with coronary heart disease were excluded from the study, different types of silent cardiomyopathies, such as hypertrophic cardiomyopathy (HCM) [53], arrhythmogenic right ventricular cardiomyopathy (ARVC) [54] or idiopathic dilated cardiomyopathy (DCM) [55] that may affect our results, had not included in the study’s exclusion criteria. Thirdly, since cardiac implantable electronic devices (CIED) are widely used as a therapeutic measure in cardiac rhythm disorders and heart failure management in the DKD population, the possible impact of exercise intervention in CIED-related complications [56] and quality of life [57] of HD patients should have been taken under consideration. However, in the current randomized clinical trial, we neither include patients with CIEDs nor examined the protentional impact on the study population considering related complications (such as infections) and quality of life. Finally, the long-term prognostic ability of exercise training was not evaluated.

## 5. Conclusions

In conclusion, a 6-month, home-based combined exercise training program can improve cardiac autonomic function, cardiorespiratory efficiency, and metabolic profile in DKD patients in HD. Encouraging DKD patients to increase their physical activity levels may be the key to improving their daily life and reducing cardiovascular risk and mortality.

## Figures and Tables

**Figure 1 life-13-00232-f001:**
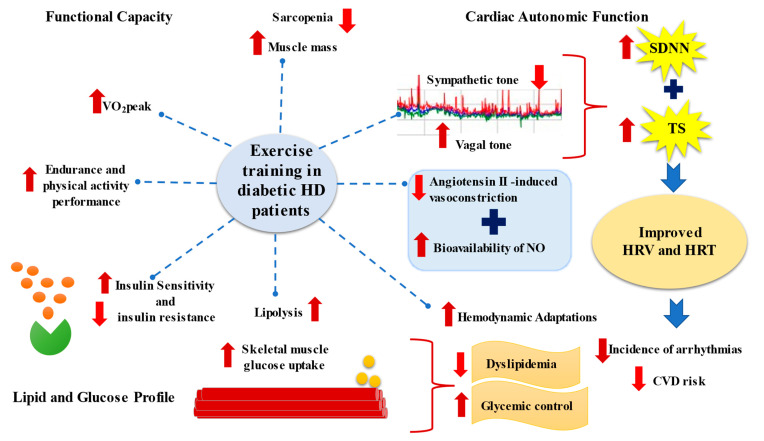
Pathophysiological mechanisms of exercise on cardiac autonomic function, functional capacity, lipid, and glucose profile on DKD patients. The arrows represent the effects of exercise.

**Figure 2 life-13-00232-f002:**
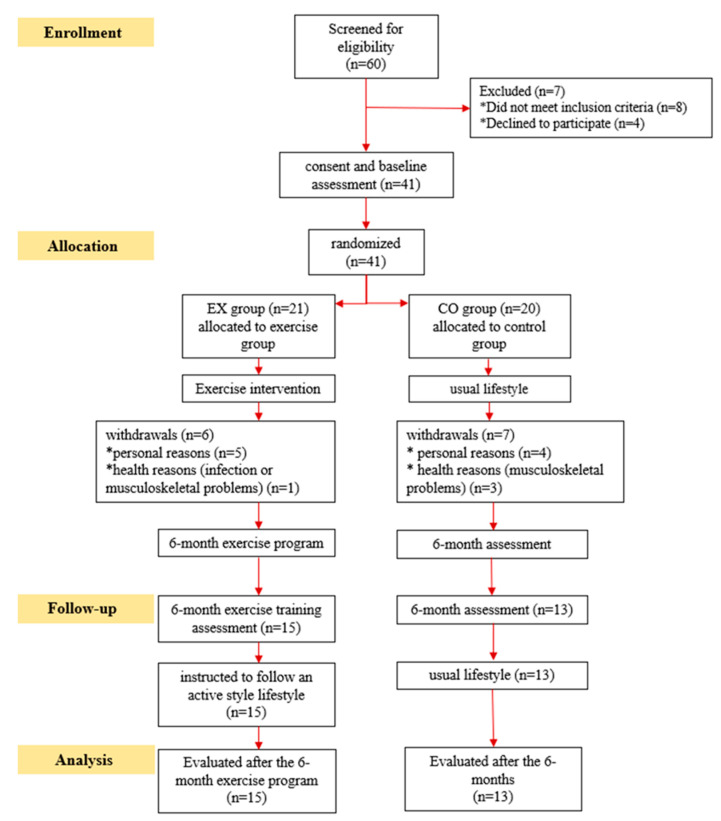
CONSORT diagram of the study design. * represent separately the reasons of withdrawals.

**Figure 3 life-13-00232-f003:**
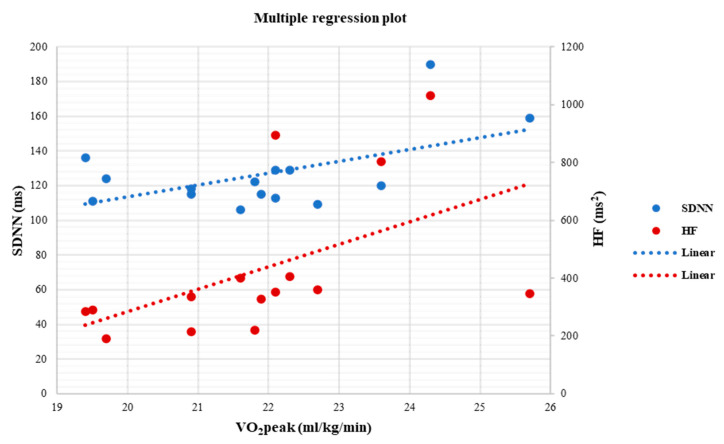
Linear regression analysis between the VO_2_peak (ml/kg/min) and SDNN (ms) (r = 0.55, *p* = 0.03) and between VO_2_peak (mL/kg/min) and HF (ms^2^) (r = 0.72, *p* < 0.05), after 6 months in the EX group.

**Figure 4 life-13-00232-f004:**
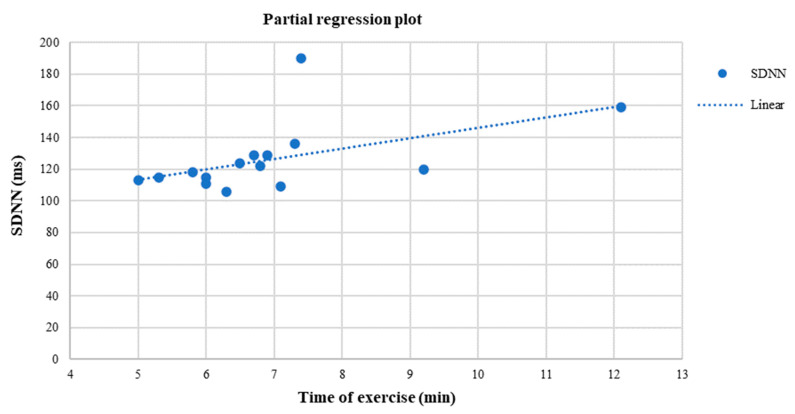
Linear regression analysis between the SDNN (ms) and time of exercise (min) (r = 0.62, *p* = 0.04) after six months in the EX group.

**Figure 5 life-13-00232-f005:**
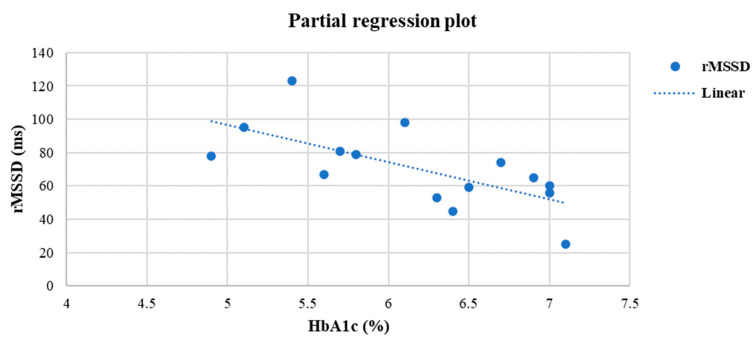
Linear regression analysis between HbA1c (%) and rMSSD (ms) after six months in the EX group (r =−0.70, *p* < 0.05).

**Table 1 life-13-00232-t001:** Patients’ medication during the study.

	EX Group	CO Group	*p*-Value
Medication at home			
Calcium channel inhibitors	6 (40.00%)	5 (38.46%)	*p* = 0.75
Antidiabetic drugs (per os)	4 (26.66%)	4 (30.76%)	*p* = 0.84
Beta-adrenergic blockers	4 (26.66%)	4 (30.76%)	*p* = 0.82
Slow and/or intermediate-acting insulin	13 (86.66%)	12 (92.30%)	*p* = 0.76
Corticosteroids	2 (13.33%)	1 (7.69%)	*p* = 0.73
Lipid-lowering drugs	8 (53.33%)	9 (69.23%)	*p* = 0.77
Diuretics	4 (26.66%)	3 (23.07%)	*p* = 0.80
Antithrombotic agents	4 (26.66%)	4 (30.76%)	*p* = 0.92
Antihypertensive agents acting on the renin-angiotensin system	10 (66.66%)	9 (69.23%)	*p* = 0.81
Other	2 (13.33%)	1 (7.69%)	*p* = 0.79
Medication during hemodialysis sessions			
Erythropoietin	14 (93.33%)	12 (92.30%)	*p* = 0.56
Levocarnitine	14 (93.33%)	13 (100.00%)	*p* = 0.85
Analogues of vitamin D, Paricalcitol	12 (80.00%)	10 (76.92%)	*p* = 0.61
Vitamin complexes	7 (46.66%)	5 (38.46%)	*p* = 0.59
Other	2 (13.33%)	3 (23.07%)	*p* = 0.82

Independent *t*-test for continuous variables. Significant at the 0.05 level (*p* < 0.05).

**Table 2 life-13-00232-t002:** Demographic and clinical characteristics of patients.

	EX Group (*n* = 15)	CO Group (*n* = 13)	*p*-Value
Sex (male/female)	10/5	7/6	*p* = 0.51
Age (years)	62.06 ± 6.34	63.30 ± 8.50	*p* = 0.66
Employment status			
Employed	3 (20.00%)	2 (15.39%)	*p* = 0.76
Unemployed	2 (13.33%)	2 (15.39%)	*p* = 0.93
Retired	10 (66.66%)	9 (69.23%)	*p* = 0.81
HD vintage (years)	6.53 ± 5.70	4.70 ± 3.17	*p* = 0.31
Dry weight (kg)	78.80 ± 17.34	79.38 ± 15.01	*p* = 0.92
BMI (kg/m^2^)	28.28 ± 6.22	29.05 ± 5.71	*p* = 0.73
Dialysis access			
Arteriovenous fistula or graft	9 (60.00%)	5 (38.46%)	*p* = 0.27
Central venous catheter	6 (40.00%)	8 (61.54%)	*p* = 0.45
Previous transplantation			
Yes	3 (20.00%)	2 (15.39%)	*p* = 0.76
No	12 (80.00%)	11 (84.61%)	*p* = 0.76
Primary causes of ESKD			
Diabetes mellitus	7 (46.66%)	6 (46.15%)	*p* = 0.77
Hypertension	5 (33.33%)	3 (23.07%)	*p* = 0.81
Glomerulonephritis	2 (13.33%)	2 (15.38%)	*p* = 0.79
Other	1 (6.66%)	2 (15.38%)	*p* = 0.73
CCΙ	7.20 ± 1.78	7.15 ± 1.67	*p* = 0.94
Comorbidities			
Hypertension	7 (46.66%)	6 (46.15%)	*p* = 0.64
Dyslipidemia	2 (13.33%)	2 (15.38%)	*p* = 0.91
Hypothyroidism	0 (0.00%)	1 (7.69%)	*p* = 0.84
Diabetic retinopathy	2 (13.33%)	1 (7.69%)	*p* = 0.71
Peripheral neuropathy	1 (6.66%)	0 (0.00%)	*p* = 0.83
Multiple myeloma	0 (0.00%	1 (7.69%)	*p* = 0.82
Secondary hyperthyroidism	1 (6.66%)	1 (7.69%)	*p* = 0.95
Venous insufficiency	2 (13.33%)	2 (15.38%)	*p* = 0.92
Mitral valve insufficiency	1 (6.66%)	0 (0.00%)	*p* = 0.82
Varicose veins	1 (6.66%)	0 (0.00%)	*p* = 0.82
Osteoporosis	0 (0.00%)	1 (7.69%)	*p* = 0.84
COPD	1 (6.66%)	0 (0.00%)	*p* = 0.82
Other	1 (6.66%)	0 (0.00%)	*p* = 0.85

ESKD: End-stage kidney disease; HD: Hemodialysis; BMI: Body mass index; CCI: Charlson Comorbidity Index; COPD: Chronic Obstructive Pulmonary Disease. Independent *t*-test for continuous variables. Significant at the 0.05 level (*p* < 0.05).

**Table 3 life-13-00232-t003:** Blood analysis at baseline and the end of the study.

	EX Group				CO Group				EX vs. CO Group			
	Baseline	After 6-Months	*p*-Value	Intra-Observer Variability ICC (95% CI)	Baseline	After 6-Months	*p*-Value	Intra-Observer Variability ICC (95% CI)	Pre	Inter-Observer Variability ICC (95% CI)	Post	Inter-Observer Variability ICC (95% CI)
Hematocrit (%)	35.86 ± 3.28	36.22 ± 3.54	*p* = 0.19	0.97 (0.93/0.99)	36.13 ± 2.39	36.35 ± 2.23	*p* = 0.31	0.78 (0.55/0.99)	*p* = 0.80	0.41 (−1.06/0.76)	*p* = 0.78	0.21 (−1.57/0.76)
Hemoglobin (g/dL)	11.61 ± 0.85	11.80 ± 1.35	*p* = 0.38	0.85 (0.56/0.95)	11.68 ± 0.68	11.73 ± 0.76	*p* = 0.55	0.90 (0.69/0.97)	*p* = 0.81	−0.75 (−4.75/0.46)	*p* = 0.73	−1.00 (−5.56/0.38)
Na+ (mg/dL)	138.46 ± 2.47	138.53 ± 3.06	*p* = 0.68	0.93 (0.77/0.97)	137.84 ± 4.77	137.85 ± 4.48	*p* = 0.94	0.93 (0.77/0.97)	*p* = 0.66	−1.11 (−5.92/0.35)	*p* = 0.61	−0.41 (−3.64/0.56)
Κ+ (mg/dL)	5.18 ± 0.71	5.16 ± 0.54	*p* = 0.71	0.87 (0.61/0.95)	5.04 ± 0.57	5.07 ± 0.55	*p* = 0.69	0.78 (0.28/0.93)	*p* = 0.57	0.04 (−2.12/0.70)	*p* = 0.52	0.41 (−0.91/0.82)
Ca+ (mg/dL)	8.81 ± 0.71	8.94 ± 0.74	*p* = 0.66	0.81 (0.44/0.93)	8.67 ± 0.83	8.52 ± 0.91	*p* = 0.27	0.95 (0.84/0.98)	*p* = 0.64	−0.60 (−4.24/0.51)	*p* = 0.65	0.009 (−2.24/0.69)
*p* (mg/dL)	4.88 ± 1.07	4.82 ± 1.03	*p* = 0.72	0.99 (0.97/0.99)	5.00 ± 0.59	4.90 ± 0.51	*p* = 0.50	0.66 (0.45/0.84)	*p* = 0.72	0.12 (−1.87/0.73)	*p* = 0.77	0.22 (−1.52/0.76)
Mg+ (mg/dL)	2.01 ± 0.14	2.04 ± 0.11	*p* = 0.54	0.94 (0.82/0.98)	2.02 ± 0.22	2.03 ± 0.18	*p* = 0.84	0.85 (0.53/0.95)	*p* = 0.89	0.35 (−1.11/0.80)	*p* = 0.95	0.58 (−0.34/0.87)
Fe+ (mg/dL)	72.73 ± 10.40	72.53 ± 10.82	*p* = 0.97	0.74 (0.63/0.98)	72.84 ± 11.48	72.92 ± 11.60	*p* = 0.76	0.82 (0.43/0.94)	*p* = 0.79	−0.14 (−2.74/0.65)	*p* = 0.44	0.53 (−0.53/0.85)
Urea (mg/dL)	136.20 ± 24.83	136.40 ± 15.20	*p* = 0.81	0.41 (−0.73/0.80)	136.84 ± 23.04	136.84 ± 20.49	*p* = 0.98	0.38 (0.16/0.79)	*p* = 0.94	0.06 (−2.05/0.71)	*p* = 0.33	−0.96 (−5.44/0.40)
Creatinine (mg/dL)	7.32 ± 1.70	7.37 ± 1.84	*p* = 0.68	0.85 (0.57/0.95)	7.54 ± 1.65	7.58 ± 1.60	*p* = 0.92	0.89 (0.64/0.96)	*p* = 0.72	0.65 (−0.13/0.89)	*p* = 0.16	0.55 (−0.46/0.86)
ALP (mg/dL)	76.13 ± 11.10	75.93 ± 17.13	*p* = 0.33	0.58 (−0.24/0.85)	75.92 ± 9.66	75.69 ± 11.10	*p* = 0.68	0.97 (0.92/0.99)	*p* = 0.95	0.59 (−0.32/0.87)	*p* = 0.68	0.47 (−0.71/0.84)
Uric acid (mg/dL)	6.21 ± 0.57	6.11 ± 0.62	*p* = 0.46	0.64 (−0.04/0.88)	6.26 ± 0.88	6.26 ± 0.92	*p* = 0.99	0.67 (−0.02/0.79)	*p* = 0.84	0.33 (−1.17/0.79)	*p* = 0.61	−0.37 (−3.51/0.58)
Serum albumin (g/dL)	5.02 ± 1.08	5.00 ± 1.03	*p* = 0.75	−0.15 (−2.44/0.61)	4.99 ± 0.74	5.00 ± 0.64	*p* = 0.91	0.79 (0.33/0.93)	*p* = 0.93	0.53 (−0.52/0.85)	*p* = 0.98	−0.02 (−2.37/0.68)
SGOT (IU/L)	15.46 ± 3.41	15.13 ± 3.66	*p* = 0.82	0.94 (0.82/0.98)	15.76 ± 6.32	15.69 ± 6.47	*p* = 0.77	0.97 (0.92/0.99)	*p* = 0.87	0.16 (−1.73/0.74)	*p* = 0.38	−1.07 (−5.79/0.36)
SGPT (IU/L)	16.40 ± 3.88	16.46 ± 3.24	*p* = 0.76	0.91 (0.73/0.97)	15.84 ± 6.44	16.15 ± 5.47	*p* = 0.23	0.96 (0.89/0.99)	*p* = 0.78	0.17 (−1.73/0.75)	*p* = 0.35	−0.32 (−3.35/0.59)
FPG (mg/dL)	144.73 ± 41.13	121.66 ± 36.28	*p* < 0.05	0.95 (0.87/0.98)	152.69 ± 60.55	151.76 ± 61.29	*p* = 0.42	0.55 (0.10/0.79)	*p* = 0.65	−1.07 (−5.79/0.36)	*p* = 0.20	−1.22 (−6.30/0.32)
TC (mg/dL)	239.40 ± 80.46	224.53 ± 84.05	*p* = 0.04	0.82 (0.65/0.94)	237.15 ± 95.69	237.15 ± 97.02	*p* = 0.98	0.87 (0.76/0.98)	*p* = 0.94	−0.05 (−2.44/0.68)	*p* = 0.79	0.08 (−2.00/0.72)
TG (mg/dL)	191.06 ± 62.57	171.66 ± 40.12	*p* = 0.01	0.92 (0.77/0.97)	179.69 ± 53.11	179.84 ± 54.30	*p* = 0.97	0.97 (0.93/0.99)	*p* = 0.61	0.57 (−0.41/0.86)	*p* = 0.53	0.30 (−1.27/0.78)
HDL (mg/dL)	43.53 ± 20.02	56.00 ± 22.60	*p* < 0.05	0.86 (0.58/0.96)	44.07 ± 19.92	43.84 ± 19.87	*p* = 0.78	0.89 (0.73/0.95)	*p* = 0.94	0.59 (0.33/0.96)	*p* = 0.01	0.86 (0.56/0.45)
LDL (mg/dL)	102.93 ± 16.78	97.06 ± 15.07	*p* = 0.06	0.58 (0.45/0.75)	94.53 ± 22.45	94.92 ± 22.82	*p* = 0.48	0.99 (0.99/0.99)	*p* = 0.26	0.89 (0.65/0.94)	*p* = 0.66	−0.10 (−2.63/0.66)
HbA1c (%)	6.85 ± 0.69	6.16 ± 0.70	*p* < 0.05	0.86 (0.60/0.95)	7.01 ± 1.20	7.04 ± 1.15	*p* = 0.53	0.93 (0.84/0.98)	*p* = 0.67	0.04 (−2.12/0.70)	*p* = 0.04	0.007 (−2.04/0.71)

EX exercise group; CO: control group; ICC: intraclass correlation coefficient; 95% CI: CI: 95% confidence interval (lower bound/upper bound); Na: Sodium; p: Potassium; Ca: Calcium; Mg: Magnesium; P: Phosphorus; Fe: Iron; ALP: Alkaline phosphatase; FPG: Fasting plasma glucose; HDL: High-density lipoprotein; LDL: Low-density lipoprotein; TC: Total cholesterol; TG: Triglycerides; HbA1c: glycated hemoglobin. Data are expressed as mean ± SD. *p* < 0.05: baseline vs. 6 months follow-up; *p* < 0.05: group EX vs. CO.

**Table 4 life-13-00232-t004:** Cardiorespiratory efficiency at baseline and the end of the study.

	EX Group				CO Group				EX vs. CO Group			
	Baseline	After 6-Months	*p*-Value	Intra-Observer Variability ICC (95% CI)	Baseline	After 6-Months	*p*-Value	Intra-Observer Variability ICC (95% CI)	Pre	Inter-Observer Variability ICC (95% CI)	Post	Inter-Observer Variability ICC (95% CI)
Time (min)	6.45 ± 2.04	6.96 ± 1.73	*p* = 0.03	0.94 (0.84/0.98)	6.40 ± 1.24	6.43 ± 1.32	*p* = 0.66	0.91(0.97/0.99)	*p* = 0.36	0.64 (−0.16/0.89)	*p* = 0.88	0.30 (−1.27/0.78)
METs (%pred)	67.20 ± 4.79	70.53 ± 4.18	*p* < 0.05	0.83 (0.50/0.94)	67.00 ± 4.37	66.92 ± 5.04	*p* = 0.89	0.94 (0.82/0.98)	*p* = 0.95	0.72 (0.009/0.91)	*p* = 0.04	0.66 (−0.10/0.89)
VO_2_peak (mL/kg/min)	19.94 ± 2.13	21.90 ± 1.75	*p* < 0.05	0.90 (0.72/0.96)	19.80 ± 1.90	19.87 ± 1.87	*p* = 0.40	0.98 (0.77/0.99)	*p* = 0.68	0.77 (0.25/0.93)	*p* < 0.05	0.76 (0.23/0.92)
RERmax	1.12 ± 0.10	1.14 ± 0.09	*p* = 0.54	0.79 (0.38/0.93)	1.09 ± 0.07	1.09 ± 0.08	*p* = 0.98	0.73 (0.44/0.98)	*p* = 0.41	0.37 (−1.04/0.81)	*p* = 0.51	0.44 (−1.01/0.78)
VO_2_/HRmax	11.81 ± 1.88	11.90 ± 1.97	*p* = 0.74	0.67 (0.21/0.88)	11.59 ± 1.32	11.64 ± 1.26	*p* = 0.50	0.96 (0.65/0.98)	*p* = 0.73	0.77 (0.26/0.93)	*p* = 0.36	0.72 (0.08/0.91)
VE/VO_2_max	31.41 ± 4.46	29.67 ± 4.42	*p* < 0.05	0.82 (0.45/0.95)	31.63 ± 5.60	31.33 ± 5.27	*p* = 0.22	0.77 (0.32/0.95)	*p* = 0.51	0.70 (0.04/0.91)	*p* = 0.24	0.75 (0.20/0.92)
VE/VCO_2_max	35.77 ± 4.96	34.32 ± 5.07	*p* = 0.01	0.76 (0.37/0.89)	36.66 ± 7.75	36.32 ± 7.33	*p* = 0.23	0.98 (0.94/0.99)	*p* = 0.14	0.77 (0.26/0.91)	*p* = 0.11	0.37 (−1.03/0.81)
HRrest (bpm)	71.80 ± 8.24	69.60 ± 9.18	*p* < 0.05	0.98 (0.96/0.99)	73.69 ± 6.43	73.69 ± 5.99	*p* = 0.99	0.97 (0.92/0.99)	*p* = 0.75	0.47 (−0.70/0.84)	*p* = 0.54	0.50 (0.01/0.89)
SBPrest (mmHg)	124.53 ± 7.54	121.46 ± 8.09	*p* = 0.03	0.87 (0.63/0.95)	125.00 ± 5.00	125.61 ± 4.99	*p* = 0.45	0.91 (0.70/0.97)	*p* = 0.31	0.27 (−1.37/0.77)	*p* = 0.04	0.75 (0.06/0.93)
DBPrest (mmHg)	73.66 ± 8.12	71.66 ± 6.45	*p* = 0.06	0.93 (0.80/0.97)	74.23 ± 4.93	74.23 ± 4.93	*p* = 0.98	0.95 (0.85/0.98)	*p* = 0.25	−0.21 (−2.98/0.62)	*p* = 0.64	0.08 (−2.00/0.72)
HRmax (bpm)	132.86 ± 15.58	139.46 ± 17.95	*p* < 0.05	0.97 (0.91/0.99)	131.46 ± 10.19	131.15 ± 8.90	*p* = 0.64	0.88 (0.75/0.97)	*p* = 0.84	0.77 (0.27/0.93)	*p* = 0.04	0.65 (−0.14/0.89)
SBPmax (mmHg)	163.33 ± 8.79	155.53 ± 12.18	*p* < 0.05	0.78 (0.34/0.92)	164.38 ± 8.89	164.23 ± 8.61	*p* = 0.93	0.96 (0.89/0.99)	*p* = 0.42	0.68 (−2.05/0.71)	*p* = 0.03	0.11 (−1.89/0.73)
DBPmax (mmHg)	73.66 ± 8.12	71.00 ± 8.28	*p* = 0.01	0.94 (0.83/0.98)	74.23 ± 6.72	74.23 ± 5.71	*p* = 0.98	0.95 (0.87/0.97)	*p* = 0.93	−0.09 (−2.59/0.66)	*p* = 0.98	−1.07 (−5.78/0.36)

EX exercise group; CO: control group; ICC: intraclass correlation coefficient; 95% CI: CI: 95% confidence interval (lower bound/upper bound); METs: Metabolic equivalents for physical activity; VO_2_peak: Maximum oxygen consumption; RER: Respiratory exchange ratio; VO_2_/HRmax: Ratio between VO_2_ and maximum heart rate; VE/VO_2_max: Ventilatory equivalents for oxygen; VE/VCO2max: Ventilatory equivalents for carbon dioxide; HR: Heart rate; SBP: Systolic blood pressure; DBP: Diastolic blood pressure. Data are expressed as mean ± SD. *p* < 0.05: baseline vs. 6 months follow-up; *p* < 0.05: group EX vs. CO.

**Table 5 life-13-00232-t005:** Results were derived from the 24-h electrocardiographic monitoring for HRV and HRT analysis at baseline and the end of the study.

EX Group					CO Group				EX vs. CO Group			
	Baseline	After 6-Months	*p*-Value	Intra-Observer Variability ICC (95% CI)	Baseline	After 6-Months	*p*-Value	Intra-Observer Variability ICC (95% CI)	Pre	Inter-Observer Variability ICC (95% CI)	Post	Inter-Observer Variability ICC (95% CI)
HRV												
HR (bpm)	74.33 ± 12.31	73.13 ± 10.80	*p* = 0.62	0.81 (0.43/0.93)	73.30 ± 6.79	73.46 ± 5.99	*p* = 0.67	0.99 (0.96/0.99)	*p* = 0.79	−0.56 (−4.12/0.52)	*p* = 0.92	−0.17 (−0.64/0.39)
TP (ms^2^)	978.33 ± 388.51	1118.84 ± 446.64	*p* = 0.04	0.90 (0.70/0.96)	934.10 ± 403.79	935.48 ± 404.48	*p* = 0.70	0.97 (0.89/0.99)	*p* = 0.77	−0.22 (−0.67/0.35)	*p* = 0.27	−0.71 (−4.62/0.47)
Mean 24-RR intervals (ms)	842.74 ± 128.65	870.76 ± 118.27	*p* = 0.19	0.91 (0.73/0.97)	846.07 ± 83.94	847.30 ± 92.49	*p* = 0.71	0.96 (0.92/0.99)	*p* = 0.93	0.04 (−0.50/0.56)	*p* = 0.56	−0.42 (−3.65/0.56)
Time domain variables												
SDNN (ms)	95.46 ± 15.02	126.40 ± 21.95	*p* < 0.05	0.51 (−0.44/0.83)	93.92 ± 37.99	94.11 ± 33.99	*p* = 0.91	0.98 (0.95/0.99)	*p* = 0.88	0.08 (−2.00/0.72)	*p* = 0.03	0.78 (0.55/0.98)
SDANN (ms)	73.60 ± 22.76	90.86 ± 27.05	*p* = 0.02	0.78 (0.36/0.92)	75.30 ± 20.54	74.61 ± 20.03	*p* = 0.70	0.97 (0.91/0.99)	*p* = 0.83	0.11 (−1.91/0.72)	*p* = 0.17	0.26 (−1.39/0.77)
rMSSD (ms)	55.06 ± 25.92	70.53 ± 23.73	*p* < 0.05	0.88 (0.66/0.96)	55.76 ± 13.26	55.30 ± 10.16	*p* = 0.72	0.97 (0.90/0.98)	*p* = 0.93	0.05 (−0.48/0.57)	*p* = 0.02	0.61 (−1.58/0.75)
pNN50 (%)	8.66 ± 7.20	12.20 ± 11.16	*p* = 0.04	0.91 (0.75/0.97)	8.07 ± 7.52	8.07 ± 7.87	*p* = 0.99	0.99 (0.97/0.99)	*p* = 0.86	0.45 (−0.80/0.83)	*p* = 0.02	0.65 (0.11/0.78)
Frequency domain variables												
VLF (ms^2^)	1460.84 ± 950.48	1550.40 ± 1331.76	*p* = 0.87	−0.02 (−1.97/0.78)	1501.31 ± 1152.81	1526.73 ± 1148.43	*p* = 0.67	0.92 (0.76/0.97)	*p* = 0.86	0.29 (−0.28/0.71)	*p* = 0.39	0.11 (−0.44/0.61)
LF (ms^2^)	155.53 ± 53.91	108.28 ± 48.04	*p* = 0.03	0.91 (0.74/0.97)	154.65 ± 69.24	154.08 ± 67.20	*p* = 0.75	0.99 (0.89/0.99)	*p* = 0.98	−0.03 (−2.39/0.68)	*p* = 0.01	0.58 (−1.78/0.69)
HF (ms^2^)	305.04 ± 169.63	430.33 ± 259.49	*p* = 0.01	0.92 (0.71/0.98)	304.90 ± 108.38	304.45 ± 109.54	*p* = 0.49	0.88 (0.33/0.98)	*p* = 0.69	0.24 (−1.47/0.76)	*p* = 0.18	0.42 (0.20/0.78)
LF (n.u.)	16.34 ± 15.70	12.63 ± 8.28	*p* = 0.02	0.65 (−0.04/0.88)	11.21 ± 5.77	11.30 ± 5.90	*p* = 0.37	0.98 (0.94/0.99)	*p* = 0.61	0.13 (−0.42/0.62)	*p* = 0.18	0.56 (0.11/0.89)
HF (n.u.)	62.89 ± 17.49	81.07 ± 42.81	*p* = 0.12	0.42 (−0.70/0.80)	62.47 ± 21.76	62.66 ± 21.69	*p* = 0.09	0.83 (0.56/0.98)	*p* = 0.95	0.39 (−0.99/0.81)	*p* = 0.72	0.08 (−2.01/0.72)
LF/HF	2.08 ± 1.42	1.91 ± 1.28	*p* = 0.26	0.26 (−1.18/0.72)	2.12 ± 2.03	2.10 ± 1.90	*p* = 0.71	0.95 (0.90/0.97)	*p* = 0.96	−0.11 (−2.66/0.65)	*p* = 0.08	−0.006 (−2.48/0.67)
HRT												
TO (%)	0.02 ± 0.01	0.00 ± 0.00	*p* = 0.17	0.001 (−1.98/0.66)	0.00 ± 0.00	0.00 ± 0.00	*p* = 0.33	0.99 (0.99/1.00)	*p* = 0.22	−0.003 (−2.28/0.69)	*p* = 0.12	0.11 (−0.44/0.60)
TS (ms/RR)	6.54 ± 4.17	8.02 ± 4.65	*p* = 0.01	0.94 (0.82/0.98)	6.69 ± 5.11	6.50 ± 4.77	*p* = 0.35	0.95 (0.84/0.98)	*p* = 0.88	−0.001 (−0.53/0.53)	*p* = 0.40	−0.001 (−2.27/0.69)
Mean 24-RR intervals (ms)	1641.49 ± 154.43	2343.42 ± 151.25	*p* = 0.11	0.47 (0.11/0.96)	1639.72 ± 104.26	1641.11 ± 105.13	*p* = 0.64	0.98 (0.96/0.99)	*p* = 0.50	0.03 (−2.16/0.70)	*p* = 0.52	0.19 (−1.62/0.75)

EX exercise group; CO: control group; ICC: intraclass correlation coefficient; 95% CI: 95% confidence interval (lower bound/upper bound);HRV: Heart rate variability; HRT: Heart rate turbulence; TP: Total; RR intervals: Time intervals between two successive heartbeats; SDNN: standard deviation of RR intervals; SDANN: Standard Deviation of the 5 min Average NN intervals; rMSSD: root mean square of successive differences between normal heartbeats; pNN50: The number of pairs of successive NN (R-R) intervals that differ by more than 50 ms; VLF: very low frequency; LF: Low frequency; HF: High frequency; ΤO: turbulence onset; TS: turbulence slope. Data are expressed as mean ± SD. *p* < 0.05: baseline vs. 6 months follow-up; *p* < 0.05: group EX vs. CO.

**Table 6 life-13-00232-t006:** Multiple regression analysis with SDNN as an independent variable, at the end of the study.

Model	B	β	*t*-Test	*p*
Participation to exercise	−6.142		−0.076	*p* = 0.94
Age	0.641	0.185	0.968	*p* = 0.36
HD vintage	−0.265	−0.069	−0.305	*p* = 0.76
Hb	0.201	0.012	0.065	*p* = 0.94
FPG	0.250	0.414	2.098	*p* = 0.06
HbA1c	−19.621	−0.652	−2.855	*p* = 0.02
VO_2_peak	8.346	0.666	3.433	*p* < 0.05
R^2^ = 0.724				
F = 3.505				

HD: Hemodialysis; Hb: Hemoglobin; FPG: Fasting plasma glucose; HbA1c: glycated hemoglobin; VO_2_peak: Maximum oxygen consumption *p* < 0.05.

**Table 7 life-13-00232-t007:** Multiple regression analysis with VO_2_peak as an independent variable, at the end of the study.

Model	B	β	*t*-Test	*p*
Participation to exercise	20.670		4373	*p* < 0.05
Age	0.017	0.063	0.261	*p* = 0.80
HD vintage	0.158	0.514	2.051	*p* = 0.08
SDNN	0.044	0.549	2.366	*p* = 0.03
pNN50	0.017	0.126	0.665	*p* = 0.53
SDANN	−0.008	−0.124	−0.401	*p* = 0.70
rMSSD	−0.043	−0.582	−2.833	*p* = 0.03
HF (ms^2^)	0.005	0.729	3.842	*p* < 0.05
LF (ms^2^)	−0.013	−0.344	−0.962	*p* = 0.37
R^2^ = 0.863F = 4.706				

HD: Hemodialysis; SDNN: standard deviation of RR intervals; pNN50: The number of pairs of successive NN (R-R) intervals that differ by more than 50 ms; rMSSD: root mean square of successive differences between normal heartbeats; SDANN: Standard Deviation of the 5 min Average NN intervals; LF: Low frequency; HF: High frequency. *p* < 0.05

## Data Availability

The data presented in this study are available on request from the corresponding author. The data are not publicly available due to ethical restrictions.

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
