# Peer review of "Effects of Home-Based Exercise Training on Cardiac Autonomic Neuropathy and Metabolic Profile in Diabetic Hemodialysis Patients"

_life, 2023, doi:10.3390/life13010232_

Round 1

Reviewer 1 Report

To:

Editorial Board

Life

Title: “Effects of home-based exercise training on cardiac autonomic neuropathy and metabolic profile in diabetic hemodialysis patients”

Dear Editor,

I read this paper and I think that:

-       The small sample size is a limitation of the study. This should be discussed in a dedicated limitation section.

-       Please provide a post-hoc sample size calculation.

-       The reproducibility of the measurements should be provided. Please provide inter and/or intraoberver variability coefficients for the measurements.

-       ALL comorbidities should be described.

-       Pharmacological treatments should be described.

-       A multivariate regression analysis should be performed in order to evaluate the impact of confounding factors on final results.

Reviewer 2 Report

This paper  aimed to investigate the effects of home-based exercise training program on Cardiac Autonomic Neuropathy and metabolic profile in Diabetic Kidney Disease patients undergoing maintenance hemodialysis. Authors found that a 6-month home-based mixed-type exercise program can improve cardiac autonomic function and metabolic profile in Diabetic Kidney Disease patients on hemodialysis. Here you find minor comments in order to improve the manuscript

While authors excluded patients with history of clinical coronary heart disease within the previous six months, they did not investigate about presence at baseline of silent cardiomyopathy (also considering  potential linked symptoms) such as HCM (DOI: 10.1016/j.ijcard.2022.03.028), ARVC (DOI: 10.1016/j.jacep.2021.12.002) or idiopatic dilated cardiomyopathy (doi: 10.1016/j.gim.2022.11.009) that may potentially impact.Please amplify this important point in discussion/limitations and cite 3 suggested references

Since cardiac implantable electronic devices (CIEDs) are today widely used, also in diabetic kidney disease patients undergoing hemodialysis, authors should better define the potential impact on study population considering related infections (DOI: 10.1097/MD.0000000000002587) as well as quality of life ( DOI: 10.1007/s40520-018-1088-5 ). Please amplify this important point in limitations and cite 2 suggested refereice

A nice figure showing how exercise could impact on cardiac autonomic function and functional capacity in Diabetic Kidney Disease patients is extremely important for readers, focusing on Pathophysiological Mechanisms

Round 2

Reviewer 1 Report

authors well addressed my previous comments. The paper improved very much

Reviewer 2 Report

Paper is now extremely complete. Congratulations since manuscript significantly improved , great work